# A Generic View Planning System Based on Formal Expression of Perception Tasks [note 1]

**DOI:** 10.3390/e24050578

**Published:** 2022-04-20

**Authors:** Yanzi Kong, Feng Zhu, Haibo Sun, Zhiyuan Lin, Qun Wang

**Affiliations:** 1Key Laboratory of Opto-Electronic Information Processing, Chinese Academy of Sciences, Shenyang 110169, China; kongyanzi@sia.cn (Y.K.); sunhaibo@sia.cn (H.S.); linzhiyuan@sia.cn (Z.L.); wangqun@sia.cn (Q.W.); 2Shenyang Institute of Automation, Chinese Academy of Sciences, Shenyang 110169, China; 3Institutes for Robotics and Intelligent Manufacturing, Chinese Academy of Sciences, Shenyang 110169, China; 4University of Chinese Academy of Sciences, Beijing 100049, China; 5Faculty of Robot Science and Engineering, Northeastern University, Shenyang 110819, China

**Keywords:** active perception, view planning, information expression, next best view, entropy reduction

## Abstract

View planning (VP) is a technique that guides the adjustment of the sensor’s postures in multi-view perception tasks. It converts the perception process into active perception, which improves the intelligence and reduces the resource consumption of the robot. We propose a generic VP system for multiple kinds of visual perception. The VP system is built on the basis of the formal description of the visual task, and the next best view is calculated by the system. When dealing with a given visual task, we can simply update its description as the input of the VP system, and obtain the defined best view in real time. Formal description of the perception task includes the task’s status, the objects’ prior information library, the visual representation status and the optimization goal. The task’s status and the visual representation status are updated when data are received at a new view. If the task’s status has not reached its goal, candidate views are sorted based on the updated visual representation status, and the next best view that can minimize the entropy of the model space is chosen as the output of the VP system. Experiments of view planning for 3D recognition and reconstruction tasks are conducted, and the result shows that our algorithm has good performance on different tasks.

## 1. Introduction

Perception is a crucial operation for robots to understand and express the surrounding environment. It is defined as a task of collecting and processing data obtained by various hardware sensors and generating real-time perception results, which can be divided into two stages, sensing and comprehension. Generally, intelligent perception means perception that employs a smart algorithm in the comprehension stage. However, we can also improve the intelligence level of the perception task in its sensing stage. This can be achieved in two ways. (a) Enhancing the sensor’s ability to feel the environment. For example, Wan et al. [1] develop a bimodal artificial sensory neuron (BASE) based on ionic/electronic hybrid neuromorphic electronics to implement visual–haptic fusion, which helps to build a highly integrated perceptual system to access massive sensory data for improving current cyborg technologies and artificial intelligence. (b) Automatically adjusting the parameters of the sensing system during the sense process based on the current perception result. This links the sensing and comprehension stages into a closed-loop system, which can improve the system’s ability to interact with uncertain environments. This concept was first proposed by Connolly in 1985 [2].

Visual perception is the perception task where vision sensors such as a visible light camera or radar are used to collect the images or point clouds of the objects. It generates a visual representation of the environment, and finally reaches the goal of the perception task. For example, the reconstruction task aims to create a full digital model of an object, and the recognition task is to identify the candidate class to which the object belongs. Since data obtained at one view are not always enough to receive the target, we need to adjust the sensors’ postures to sense the environment at multiple views. Visual sensors installed on the robotic platforms explore the circumstances, following the movement of their backbones. Traditionally, the movement of the sensors is set before perception, so it is a passive way to obtain information. We focus on improving the self-adjustment ability of the perception system, as (b) in the previous paragraph. View planning (VP) is employed for the multi-view tasks. It automatically adjusts the postures of sensors, to improve the efficiency of perception based on the prior information and the observed data [3]. We call the perception task with VP active perception.

For an active perception task, the next best view (NBV) is determined according to several factors of the VP system, including the goal of the perception task, the capabilities of the sensor system, the perceived objects, the optimization goal of view planning and the status of the perception result. The existing literature is designed for a particular task. For example, Chen et al. [4] proposed a VP algorithm for 3D reconstruction. They used the ductility of the smooth surface of the object to predict the curve of an unknown area, and then calculated the NBV by the predicted curve. Wu et al. [5] provided 3D ShapeNets to generate the NBV for recognition. In these studies, the factors are fixed in the VP algorithm, which prevents the algorithm from being extended to other perception tasks.

With the existing algorithms, when we deal with a new perception task, the VP system needs to be redesigned, and there is a big price to pay for doing this. To deal with this problem, we are committed to building a framework of a VP system that can handle different active visual perception tasks. The structure of the generic VP system is shown in Figure 1. When faced with a specific perception task, it only needs to formally describe the task as input to the view planning module according to the system requirements. Part of the status of the system will be set before a perception task, and others will be autonomously updated with the received data during the iterate perception. The NBV is planned by the view planning module based on the formal description of the task. The proposed system can improve the efficiency and reduce the energy consumption of robots during the visual perception process. Our contributions are listed as follows:A generic view planning system is proposed. It is suitable for multiple visual tasks with a single object in the scene.We define a series of statuses that can express the key elements of different perception tasks, and describe these statuses in a formal way.Statuses are initialized before a given perception task, and they are updated to represent the real-time state of the VP system. The NBV is chosen based on the current state of the system.

Experiments on reconstruction tasks and recognition tasks were executed, which proved the effectiveness of our algorithm for different tasks. The rest of this paper is organized as follows: the second section reviews the literature. The third section introduces our generic VP algorithm. It explains the formal description of tasks, the update method of each status and the NBV calculation method. Experiments are represented in the fourth section with the discussion. Conclusions are given in the last section.

## 2. Related Work

To our knowledge, there is no framework that is specifically designed for view planning in different kinds of visual perception tasks. Active reconstruction and active recognition are the two scenarios that have received the most attention with the view planning process. Kriegel et al. [6] proposed a VP method for active scene exploration. It improves the speed of reconstruction and recognition. However, this VP method actually acts on active reconstruction. When it is applied independently to 3D recognition tasks, the improvement in the efficiency could not fulfil expectations.

The optimization goal of VP is to cover more surface with fewer views in reconstruction tasks. As the global optimization is an NP-hard problem, the greedy algorithm is usually employed instead of the global optimization. The view that can minimize the spatial uncertainty is selected as the NBV. When the objects’ models are known before reconstruction, the NBV is calculated based on the known surface information. VP methods that deal with the above reconstruction tasks are called model-based methods [7,8]. Scott W R. [9] used the greedy method to obtain the NBV, which can be used to explore more of the unknown surface of the object. Chen et al. [10] proposed a view map generation method based on a genetic algorithm and the min–max criterion, which achieves the goal of covering the target surface with a smaller view set, and they traversed the view set with the shortest path by Christopher’s algorithm. Kaba et al. [11] solved the model-based reconstruction problem by a non-greedy algorithm. The parameters in the evaluation function are modified at different perception statuses, and they are learned by a reinforcement learning method.

Since the information is complete enough for model-based methods, which results in a small research space, VP methods used for non-model reconstruction have received more attention. Surface-based methods analyze the surface trend to make sure that the entropy of the unknown model space will decrease the most after the detection at the next view. Chen et al. [4] utilized a point cloud to represent the model of the object. The boundary points are ranked by an evaluation function and the search direction of the top-ranked point is calculated to find the NBV. However, since only the smooth surface has a continuation trend, the surface-based methods are only suitable for the objects without rapid changes in curvature. Search-based methods make use of ray casting to calculate the information gain of each view [12,13,14,15]. They evaluate each candidate view by accumulating the gain of voxels that were passed through by rays of the camera. In these papers, voxel gain is defined in different ways. In general, the authors assign higher gains to unknown voxels that are close to the measured surface, because they consider that there is a higher probability of the presence of the surface at these voxels’ locations, making it easier to reduce the uncertainty of the unknown space. This assumption is based on general knowledge of all objects, but for some objects, they have special prior knowledge. The rough shape of an object can be predicted by the use of its partial prior information, which provides more reliability for NBV determination. To make full use of the prior information, semi-model methods are proposed by Wei et al. [16] and Kong et al. [17]. They describe the partial prior information of the object, and plan the NBV by invoking the formal prior information.

For active recognition tasks, the VP method is used to find the view that can cover the features that minimize the ambiguity of the object. As concluded in some reviews for the active recognition methods [18,19], the information gain-based, aspect graph-based and the machine learning methods are the mainstream methods to deal with the VP issues in recognition tasks. The information gain-based methods generate a set of hypotheses based on the current data, and choose the view that can maximally disambiguate the initial set of hypotheses. For example, Potthast et al. [20] choose the candidate view that may receive more features as the NBV. The relationship between candidate views and their reachable features must be known for this kind of method. The aspect graph-based methods build an aspect graph of the candidate geometries, and they use the reasoning method to uniquely determine the object’s type. Roy et al. [21] established an aspect graph as the knowledge system and inferred the NBV with a search tree. Dickinson et al. [22] proposed an aspect hierarchy to express the candidate geometries, and they chose the NBV based on the Bayesian network. In the situation where the scale of aspects or features is too large to be built as a search tree, learning methods are proposed to calculate the NBV. Wu et al. [5] represent a convolutional deep belief network to represent geometric 3D shapes as probability distributions of binary variables on a 3D voxel grid. When the recovery result is uncertain, they generate the observable images of the candidate views, and the distinctive view is chosen as the NBV. Johns et al. [23] decompose an image sequence into a set of image pairs, and select the view that achieves the maximum classification accuracy associated with the current view as the NBV. Sun et al. [24] employed a reinforcement learning method to obtain a globally optimal view.

As we focus on regular objects, whose prior information can be expressed formally, the deep learning methods are too expensive for us on large datasets. For the purpose of building a generic VP system for different kinds of tasks, the formal description method in our approach draws inspiration from the semi-model methods for reconstruction [17], which describe the structures of objects. It also draws on the active recognition method based on an aspect graph [21], which describes the status of features used to distinguish different categories of objects.

## 3. Method

The prior information of a task contains the ability of the sensor, which describes the sensor’s capabilities to sense the environment. It also contains some deterministic information about the object, which can be used to limit the category and shape of the incomplete object. We describe the prior information of objects from the feature level. Each feature contains several local surfaces with the same kinds of characteristics. The presence of features is used to distinguish candidate classes, and the attributes of features determine the rough shape of an instance. As is shown in Figure 2, $f_{2}$ only exists in the prism and $f_{3}$ only exists in the pyramid. When one of them is detected, we can determine whether the object is a prism or pyramid. The attributes of $f_{1}$ are vertices arranged counterclockwise. The attributes of $f_{2}$ are four vertices of a rectangle, and the edge of the rectangle begins with a bottom edge. The attributes of $f_{3}$ are three vertices of a triangle, and the first vertex is the apex of the pyramid. If the object’s category has been determined as a pyramid, vertices of $f_{1}$ are obtained, a local surface of $f_{3}$ is detected, and then the rough shape of the object can be predicted.

The task’s completion status is defined, which reflects the progress of an ongoing perception task. The expected completion status is given for a specific task, and the real-time completion status is updated during the perception. When the real-time completion status reaches the expected one, it means a termination of the VP procedure. With the acquisition of data in the iterative perception process, the state of the perceived space will change, which describes the data acquired during the entire perception process and is used to calculate the NBV. We describe the perception result by the visual representation status in two aspects. One is from the perspective of perceived space, which includes the voxel status, feature status and the candidate class status (“candidate status” for short). They describe the perceived space from concrete to abstract. The other one is from the perspective of candidate views, which includes the view status. It describes the information that can be observed from each candidate view. As a summary, the formal description of a visual task includes the formal expression of the task’s completion status, the prior information of the perceived object and the visual representation status.

Figure 3 illustrates the update flow chart of each state. The relevant descriptions need to be initialized before a given perception task. During the iterative measurements, the voxel status and feature status are renewed when fresh data are received. The candidate status will be changed when some classes are excluded from the candidate library according to the feature status, and the feature status is affected by the candidate status in turn. Unknown features are predicted based on the prior information library. The voxel status is renewed when the voxel is at the location of the predicted features. If a feature is detected to be nonexistent based on the voxel status, the feature status and the candidate status will be updated again. The view status is updated with the voxel status, based on the given optimal function. The completion status is calculated with the view status and the candidate status. The algorithm is terminated when the prospective task status is reached. Otherwise, the NBV will be determined on the basis of the view status, and the robot will move to the NBV for the next sense. Details of the formal description and status update of perception tasks are introduced below.

### 3.1. Formal Description of Tasks

The task’s completion status $s_{cmp}$ is described in two aspects, the class confidence status and the surface integrity status.
$$s_{cmp}:\left(s_{cfd},s_{ail}\right)$$
where $s_{cfd}$ is the class confidence state, which is utilized to indicate if the object belongs to a certain class with an ideal probability. $s_{ail}$ is a state that shows whether the available surface information about the model is left.

The prior information library contains knowledge about the sensor and the object of a task. For the description of sensors’ ability, we simply need to take the parameters of the camera into the prior information library, such as the field of view, depth of field, etc. For the prior information of the object, it includes the features’ existence in each candidate class, description of these features and the positional relationship between features. The first part is to determine which class the object belongs to; the second part is to detect the latent features, and the last part is to infer the unknown surface of an object.

For a perception task, the candidate library is built by all geometry classes that the object may belong to. The library $S_{class}$ is described as follows:$$S_{class}:\left(class_{1},\cdots class_{m},\cdots class_{M}\right)$$

There are *M* classes in the library. We add all the features, which can distinguish the candidate classes and define the rough shape of an object, into the feature set. The feature set is noted as $S_{f}$, which has *I* members. Each candidate class $class_{m}$ is defined by the presence of features in $S_{f}$.
$$class_{m}:\left(ext_{1}^{m}\cdots ext_{i}^{m}\cdots ext_{I}^{m}\right)$$

$ext_{i}^{m}$ indicates the presence of the feature $f_{i}$ in $class_{m}$. If the feature exists in this class, $ext_{i}^{m}$ is assigned a value of 1; otherwise, it is set to 0.

We simplify the local surfaces of features into polygons to express their rough shapes. Prior information about features promotes the acquisition of an object’s surfaces. According to the degree of the information’s certainty, we divide the information into three levels, including definite information, strong information and weak information. The definite information is actually the detection method of a feature. Strong and weak information are prediction methods of unknown features by the positional relationships between features. The positional relationship of two local surfaces can be defined by their common edges and the angle between them. For two features, if only the positional relationship between them is known, the relationship is called the weak information, while if the locations between the common edge and other edges in surfaces are known as well, the information is called the strong prior information.

In order to facilitate the detection and inference of features, we unify the description of all kinds of prior information into the same expression. Inspired by [17], we express the prior information by rules. There are K rules in the rule set, each of which is represented as:$$rule_{k}:\left\{S_{k}^{ant}\to f_{k}^{cons},proof_{k}\left(\right),cal_{k}\left(\right),pred_{k}\left(\right)\right\}$$

$rule_{k}$ is the information that helps us to obtain the surface points of feature $f_{k}^{cons}$. If it describes definite information, we set $S_{k}^{ant}$ as an empty one, and $proof_{k}\left(\right)$ stores the detection function to check the existence of the feature $f_{k}^{cons}$. Its input is the obtained RGBD images, and the output is a Boolean value that indicates whether the feature is detected. $f_{k}^{cons}$’s attributes are calculated by $cal_{k}\left(\right)$ and stored in the attributes set $S_{a}$. $pred_{k}\left(\right)$ is a void function that has no operation.

When we describe the strong or weak information about position relationships between the consequence feature $f_{k}^{cons}$ and several antecedent features, $S_{k}^{ant}$ contains the antecedent features, $f_{k1}^{ant}∼f_{kR}^{ant}$. The rule can only be used when the consequence is unknown and all antecedent features in $S_{k}^{ant}$ have been detected. $proof_{k}\left(\right)$ stores the detection function to check if the feature that has the given relationship with the antecedent features exists. $cal_{k}\left(\right)$ has the same function as that in rules with definite information. $pred_{k}\left(\right)$ stores the function that generates the predicted points of $f_{k}^{cons}$.

We give an example in Figure 4. In this situation, $f_{1}$ and $f_{3}$ are detected, and their vertices and edges are calculated by the definite rules about them:$$\left\{\phi \to f_{1},det\_f_{1}\left(\right),cal\_f_{1}\left(\right),pred\_void\left(\right)\right\}$$
$$\left\{\phi \to f_{1},det\_f_{3}\left(\right),cal\_f_{3}\left(\right),pred\_void\left(\right)\right\}$$

The common edge of $f_{1}$ and $f_{2}$ is known as $E(v_{4}v_{5})$, the angle between them is θ, and the relationships of the common edge and other edges in $f_{2}$ are known before. Then vertices of $f_{2}$ can be calculated, and surface points of $f_{2}$ are generated by uniform sampling of the polygon in $pred_{k}\left(\right)$ of strong rules:$$\left\{f_{1}\to f_{2},det\_f_{1}\_f_{2}\left(\right),cal\_f_{2}\left(\right),pred\_f_{1}\_f_{2}\left(\right)\right\}$$

The common edge of $f_{1}$ and $f_{4}$ is known as $E(v_{2}v_{3})$, and the common edge between $f_{3}$ and $f_{4}$ is known as $E(v_{2}v_{8})$. Then, unknown edges of $f_{4}$ are predicted to make up a square $S(v_{3}v_{8}v_{12}v_{13})$ together with the line $\bar{v_{3}v_{8}}$, which connects the start point and the end point of the detected common edges. This prior information is stored in the weak rule:$$\left\{f_{1},f_{3}\to f_{4},det\_f_{1}f_{3}\_f_{4}\left(\right),cal\_f_{4}\left(\right),pred\_f_{1}f_{3}\_f_{4}\left(\right)\right\}$$

The visual representation status includes the voxel status, the feature status, the candidate status and the view status.

We use octomap [25] to represent the model space. States of voxel nodes in the model space are divided into four types, as Figure 5 shows. The state of voxel $v_{n}$ refers to $ps_{n}$, which is initialized as “unknown”. With further observation, it turns to “occupied” or “free”. $f_{ij}$ is the *j*th local area of the feature $f_{i}$. For the unknown voxels where predicted areas are located, we give them a label of “preOccupied”, and the serial numbers of their corresponding features are tagged on them. −1, 1, 0, 2 are used to represent the “unknown”, “occupied”, “free” and “preOccupied” states, respectively.

Based on the observed data, a feature’s existence state in the perceived object can be calculated by the stored prior information. The existence state of $f_{i}$ is called $es(f_{i})$, $es_{i}$ for short. All features’ states form an array $S_{es}$, which is called the feature status. Members of $S_{es}$ are set to −1 before perception, and $es_{i}$ will turn to 0, 1, 2 or 3 with the acquisition of perceived data. Moreover, 2 is an interim state of a feature, which means that the feature exists but still needs to be observed to confirm its attributes. Meanwhile, 0, 1 and 3 are the final states of a feature. If the feature is predicted, its will be added with a prediction state, which is noted as 4. The attribute set $S_{A}$ contains the attributes of all sub-features.

**−1** unknown state, where we cannot determine whether the feature is existent or not;**0** nonexistent by detection, which means that the potential location of the feature has been detected, but this feature is still not found;**1** existent by detection, which means that the feature has already been detected from the obtained data;**2** existent by reasoning, which is the signal that the feature is existent by reasoning the candidate library;**3** nonexistent by reasoning, which is a state that the feature is nonexistent by reasoning the candidate library;**4** predicted, which means that the potential location of the feature is predicted.

Candidate status $S_{cand}$ is a vector that shows whether each candidate class in $S_{class}$ is still a possible category of the object.
$$S_{cand}:\left(cs_{1},\cdots cs_{m},\cdots cs_{M}\right)$$
where the element $cs_{m}$ is initialized to true before the perception task. When the feature status is calculated iteratively, $class_{m}$ may be excluded from the candidate classes, and its corresponding status $cs_{m}$ changes to false.

Candidate view set $S_{v}$ is generated uniformly around the object space before perception tasks. For each view $v_{c}$ in $S_{v}$, its state $vs_{c}$ reflects the information gain that can be received when the sensor moves to it as the next view. The state is calculated by the optimization function that is defined manually.

### 3.2. Status Update

The feature status is updated after each measurement. The transformation between existence states of a feature is shown in Figure 6. The received data are put into the detection functions in the prior information library to analyze the feature’s existence. $es_{i}$ is set to 1 when $req_{exd}$ is met, which means that the feature has been detected. Then, local areas of $f_{i}$ are extracted and their attributes are calculated and stored in the attribute set.
(1)$$req_{exd}:\;\exists k,k\in \left[1,K\right],\left(f_{k}^{cons}=f_{i}\right)\&\left(proof_{k}\left(\right)=1\right)$$

If the acquired information is fully used, and $f_{i}$ is still in non-final state, then $es_{i}$ is attached to a prediction state 4 when $req_{pre}$ is met, which means that the feature’s location can be inferred by its related features.
(2)$$\begin{array}{cc} req_{pre}:\;!req_{exd} & \&(\exists k,k\in \left[1,K\right],f_{k}^{cons}=f_{i})\\ \; & \&(\forall f_{kj}^{ant},f_{kj}^{ant}\in S_{k}^{ant},es\left(f_{kj}^{ant}\right)=1) \end{array}$$

$es_{i}$ is set to 0 when the detected proportion of $f_{i}$’s predicted voxels has exceeded the threshold and the feature still has not been detected. The requirement $req_{ned}$ is inferred from the voxel status.
(3)$$req_{ned}:\;\frac{1}{|PV_{i}|}\sum_{pv_{i}\in PV_{i}}U(\xi \;-\min_{p\in PC_{t}}||p-pv_{i}{\left|\right|}_{2})\geq \xi_{ned}$$
where *U* is the Heaviside step function, $PV_{i}$ is the predicted voxels of $f_{i}$, $PC_{t}$ is the point cloud observed until time *t*, ξ is a distance threshold and $\xi_{ned}$ is a percentage threshold.

The candidate library is employed to verify if $es_{i}$ should be renewed from −1 to 2. The conversion condition $req_{exr}$ is satisfied if $f_{i}$ exists in every class.
(4)$$req_{exr}:\;\forall m,m\in \left[1,M\right],ext_{i}^{m}=1$$

When $f_{i}$ is nonexistent in every candidate class, $es_{i}$ will be renewed from −1 to 3. The condition $req_{ner}$ is described as:(5)$$req_{ner}:\;\forall m,m\in \left[1,M\right],ext_{i}^{m}=0$$

The voxel status is updated when a new observed point cloud is transformed and inserted into the model space. If an unknown voxel is located on the detected surface, its state converts to “occupied”. The voxel’s state will convert to “free” when it is passed through by rays. $ps_{n}$ is set to “preOccupied” when $v_{n}$ is on the location of predicted local surfaces.
(6)$$ps_{n}=\left\{\begin{array}{cc} 2, & \sum_{i=1}^{I}U(\xi \;-\min_{pv_{i}\in PV_{i}}\left|\right|v_{n}-pv_{i}{\left|\right|}_{2})>0\\ 0, & otherwise \end{array}\right.$$

After the feature $f_{i}$ obtains its final state through detection, the candidate status will be updated with the feature status. $cs_{m}$ will turn to false when $class_{m}$ conflicts with the feature status $S_{es}^{t}$ at time *t*. The exclusion requirement $req_{exc}$ is:(7)$$\begin{array}{cc} req_{exc}:\; & P(class_{m}|S_{es}^{t})\\ \; & =\frac{P\left(S_{es}^{t}|class_{m}\right)P\left(class_{m}\right)}{\sum_{j=1}^{M}P\left(S_{es}^{t}|class_{j}\right)P\left(class_{j}\right)}=0 \end{array}$$
$P(class_{j})$ is the prior probability of $class_{j}$. At time *t*, if there are some features whose existence states are final states and are the opposite of the relevant presence states in $class_{j}$, then the probability $P(S_{es}^{t}|class_{j})$ is 0; otherwise, it is 1.

The view status is updated when the voxel status update has been done. When views have been reached, they have no contribution to the perception task any more, so their states are set to 0. Other views’ states are calculated by the evaluation function:(8)$$vs_{c}=G_{ovlp}\left(v_{c}\right)\cdot \left(G_{fd}\left(v_{c}\right)+W\cdot G_{so}\left(v_{c}\right)\right)$$

The evaluation function takes three factors into account, consisting of the overlap constraint, the feature detection and the unknown surface observation. The first component $G_{ovlp}\left(v_{c}\right)$ is to guarantee an overlap between the new and the existing point cloud for registration [13]. The feature detection factor $G_{fd}\left(v_{c}\right)$ is the kernel of the function to detect the features with higher importance, for the reason that the view with a high value at this part can not only speed up the process of recognition, but also promote the acquisition of a complete model.
(9)$$G_{fd}\left(v_{c}\right)=\max_{i\in [i,I],j\in [1,J]}w_{i}\cdot \frac{n^{ij}\left(v_{c}\right)}{n_{t}^{ij}}$$
where $w_{i}$ is a parameter that reflects the contribution of $f_{i}$ to identify the object. If the existence state of $f_{i}$ is −1, its weight $w_{i}$ is defined as the conditional entropy. If $es_{i}$ is 2, which indicates that $f_{i}$ is of no use to the identification but contributes to the object reconstruction, we give $w_{i}$ a value smaller than the minimum weight of all features that have an existence state of −1. $w_{i}$ is set to 0 otherwise. All predicted voxels of $f_{ij}$ form a voxel set, $n_{t}^{ij}$ is the number of voxels in the set, and $n^{ij}\left(v_{c}\right)$ is the number of reachable voxels at view *v* in the set.

$G_{so}\left(v_{c}\right)$ provides assistance to $G_{fd}\left(v_{c}\right)$, in the situation where no preoccupied voxel exists and the model is not complete. It is assigned by counting up all reachable unknown voxels behind the surface of the model. *W* is set to 1 when the result of $G_{fd}\left(v_{c}\right)$ is 0 for all candidate views.

$s_{cfd}$ is calculated and it will be set when the maximum confidence of all species exceeds the set threshold; otherwise, it is false. $\xi_{cfd}$ is the confidence threshold.
(10)$$s_{cfd}=\left\{\begin{array}{cc} 1, & \max_{m=1}^{M}P\left(class_{m}|S_{es}^{t}\right)\geq \xi_{cfd}\\ 0, & otherwise \end{array}\right.$$

The state $s_{ail}$ will turn to 1 from the default 0 when all candidate views could not receive available information from the environment.
(11)$$s_{ail}=\left\{\begin{array}{cc} 1, & \forall v,G(v)=0\\ 0, & otherwise \end{array}\right.$$

When $s_{cmp}$ has obtained its expected values, the perception task will come to an end. Otherwise, the view whose state has the maximum value is selected as the NBV to obtain the next measurement.
(12)$$NBV=\underset{v_{c}\in S_{v}}{\arg\max}vs_{c}$$

## 4. Experiments

To prove the efficiency of our methods for different kinds of perception tasks, experiments for reconstruction tasks and recognition tasks are performed in a simulated environment. The robotic operating system [26] is used for data transmission and virtual camera controlling. Deformed prisms and pyramids are employed as objects to be reconstructed, and 12 tanks that can be identified by 7 features are used for recognition tasks [27]. In order to focus on the VP process, we choose noticeable colors and textures as the information to detect features. The positions of the candidate views are evenly distributed on the sphere surrounding the model space, and the camera’s optical axis always points to the center of the sphere. For each experiment, we select one of the candidate views as the initial view randomly, and execute the VP method until the termination is reached. The maximum iteration is set to 20. As we expect to complete a perception task with fewer views, the number of views requested for complete visual tasks is an indicator to evaluate the efficiency of the VP algorithm. Therefore, we record the number of views required to complete each task and compare it with that for other methods to verify the effectiveness of our algorithm.

### 4.1. Experiment for Reconstruction

Experiments on reconstruction tasks were executed with the simulated objects. The candidate library and the prior information about features are generated as in Figure 7. There are 5 features in the feature set, $f_{1}∼f_{5}$, including the bottom of the object, the side of the prism, the side of the pyramid, the rough shape of the prism and the rough shape of the pyramid. The vector representing the existence states of features for the prism is (1, 1, 0, 1, 0), while it is (1, 0, 1, 0, 1) for the pyramid. $bottom_{i}$ is the sub-feature of $f_{1}$ defined by the vertices circling the bottom counterclockwise and the normal of the bottom. $side_{n}^{1}$ is one side of the prism, which is a sub-feature of $f_{2}$. Its attributes are four vertices circling it clockwise and the normal of the side. $side_{n}^{2}$ is a sub-feature of $f_{3}$. It is a side of the pyramid. $f_{3}$ has three vertices, and it is the only difference between attributes of $f_{2}$ and $f_{3}$. $rs^{1}$ is the rough shape of the prism, defined by the attributes of the bottom and the height of the prism. $rs^{2}$ is the rough shape of the pyramid, which has the same parameters as $rs^{1}$, except for the substitution of the height to the apex of the pyramid. $r_{1}∼r_{3}$ stockpile descriptions of $f_{1}∼f_{3}$ and $r_{4}∼r_{9}$ store relationships between $f_{1}∼f_{5}$.

(0, 1) is the target for completion status. ξ is set to 0.01; $\xi_{ned}$ and $\xi_{cfd}$ are set to 0.99. Eight other methods were used as control groups, including seven methods compared by Delmerico et al. [28] and the two-step active reconstruction method (T-S) proposed by Kong et al. [17]. The reference methods do not take the overlap constraint into consideration, which will cause a decrease in the required views, so we executed the experiments with (GNO) and without (GN) the overlap constraint separately to evaluate the efficiency of our algorithm on reconstruction tasks.

### 4.2. Experiment for Recognition

We conducted experiments to plan the NBV in tank recognition. The candidate library is shown in Figure 8. There are 7 features in the feature set. Each feature has only one sub-feature and they are distributed on the surface of the tank. The sub-feature $QRcode_{i}$ is located in a box. Rules $r_{1}^{2}∼r_{7}^{2}$ store description methods of each feature, while $r_{8}^{2}∼r_{49}^{2}$ are relationships between features.

(1,0) is the target for completion status. ξ is set to 0.03, $\xi_{ned}$ is set to 0.95, $\xi_{cfd}$ is set to 0.99. For each model, we tested our method against the random method and the Prior Feature Distribution Table method (PFDT). The NBV is stochastically specified among the candidate views by the random method. In the PFDT method, the features correspond to the candidate views one-to-one, and the NBV is determined according to the relationship between features.

### 4.3. Results and Discussion

The number of views required for each active reconstruction is recorded and shown in Table 1. We find that our algorithm can complete each reconstruction with a relatively small and steady number of views. According to Figure 9, which represents the average views among all models for each method, it is shown that the GN, GNO method and the T-S method have distinct advantages over other algorithms. Among them, the proposed GN method uses the lowest average number of views. When the overlap constraint is taken into consideration, the efficiency of VP is slightly affected. This result is reflected in the figure, where the average number of views needed by the GNO method is slightly higher than that by the GN method, and levels with that by the T-S method. This is caused by the overlap constraint that limits the size of unknown surface obtained from a new view. However, the overlap constraint is necessary in the real world, for the reason that the robot movement will have errors that need to be eliminated by point cloud registration.

For the active recognition tasks, the number of views needed for each tank is shown in Figure 10. The random method always needs more views for the recognition of each object. It shows that the efficiency of recognition can be promoted by a reasonable view planning method. Our method requires fewer views than the other two methods in most cases, which proves that our method is suitable for recognition tasks as well. The average number of views required by our method is less than that by the PFDT method. The reason for this result may be that the PFDT method only focuses on the area of one feature at a view, which makes the probability of other features being detected very low, and the discriminative features can only be detected view-by-view.

Through these experimental results, we can find that the proposed method plans the NBV effectively in different perception tasks, where the perception goals and measured objects are different. Furthermore, the efficiency of the generic VP is relatively higher than other methods. The results show that our method can be applied to increase the intelligence of robots or unmanned platforms. When the robot equipped with our VP system is assigned the task of multi-view visual perception, it can automatically choose the next view with the greatest contribution to the overall task. These view sequences are smaller than that selected randomly or by other methods, so that the control system of the robot does not need to sense and process the data of the environment frequently, which reduces the consumption of movement and calculation. In addition, for some exploration tasks, the smaller the number of views that need to be detected, the less the disturbance to the target in the environment.

## 5. Conclusions

We propose a generic view planning method that is suitable for multiple visual tasks. A formal description of tasks, including the prior information library, the visual representation status and the completion status, is raised to build the basis of VP for a specific task. A task begins with the initialization of its descriptions. The states are updated after each measurement. The NBV is chosen based on the current task status, until the task is ended with a signal that the completion status has received its target. Experiments for reconstruction and recognition were executed and the results show that our algorithm is effective for NBV planning in multi-tasks. However, our method will be limited when the feature’s local surfaces are too complex to be expressed with parameters, for the reason that the information of a complex object’s features is too difficult to be described manually. A deep neural net has the ability to autonomously learn the features of objects by adjusting its weights through backpropagating the error of samples in the large dataset. In the future, we will consider using deep learning methods to solve the prediction problem of complex surfaces. Moreover, as the Hebbian synapse-based reinforcement learning [29] can continuously adjust the weight between nodes in an unsupervised manner, it also provides us with a novel mean of generating the NBV end-to-end.

## Figures and Tables

**Figure 1 entropy-24-00578-f001:**
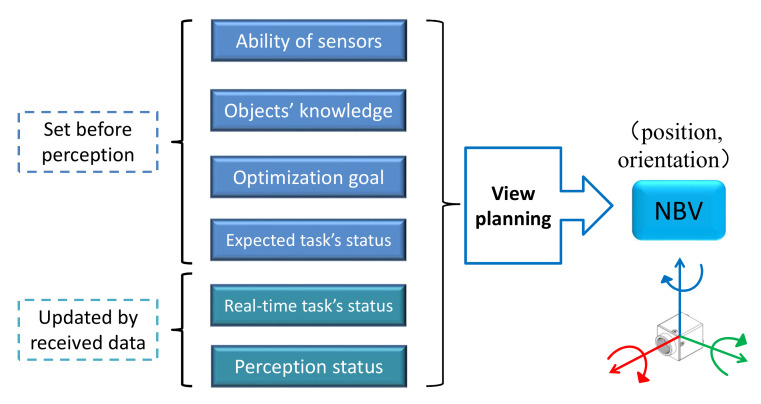
Structure of the generic view planning system. Ability of sensors, objects’ prior knowledge, optimization goal of view planning and the expected status of a perception task are set before the perception process. With data receiving at the calculated views, real-time task status and the status of perception result are updated. View planning is based on these formal descriptions and the NBV is determined for the next iteration of data collection.

**Figure 2 entropy-24-00578-f002:**
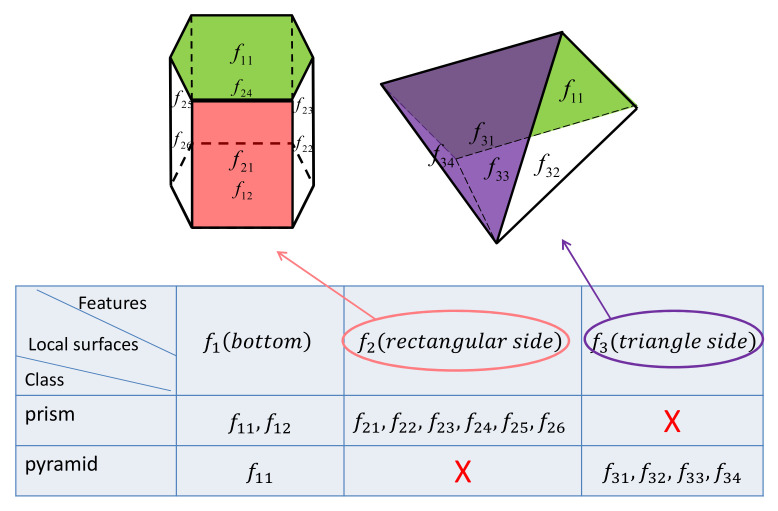
Features of different classes of objects. The prism has features $f_{1}$ and $f_{2}$, while the pyramid has features $f_{1}$ and $f_{3}$. The rough shape of an instance can be defined by the parameters of its features.

**Figure 3 entropy-24-00578-f003:**
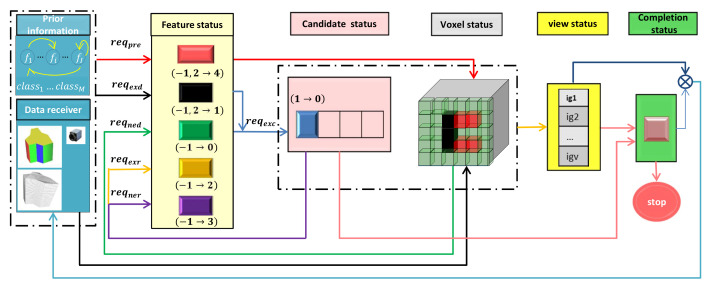
The update flow chart of each state. Boxes are basic description modules of our system and arrows represent the data streams when a new measurement is received.

**Figure 4 entropy-24-00578-f004:**
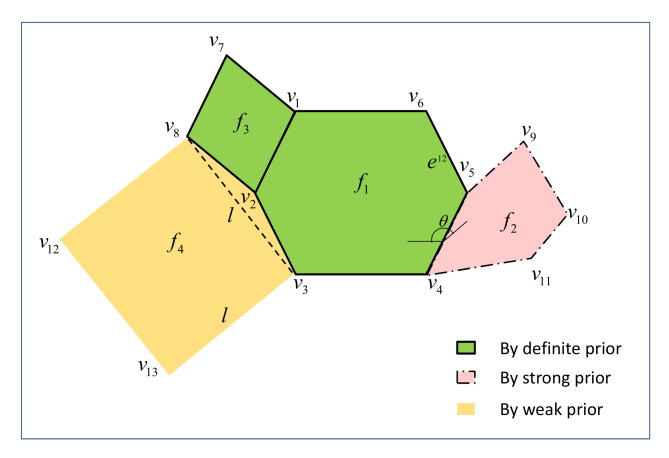
Positional relationship of local features. The green surfaces are detected by definite prior information, the pink one is predicted by strong prior information, and the yellow is predicted by weak prior information.

**Figure 5 entropy-24-00578-f005:**
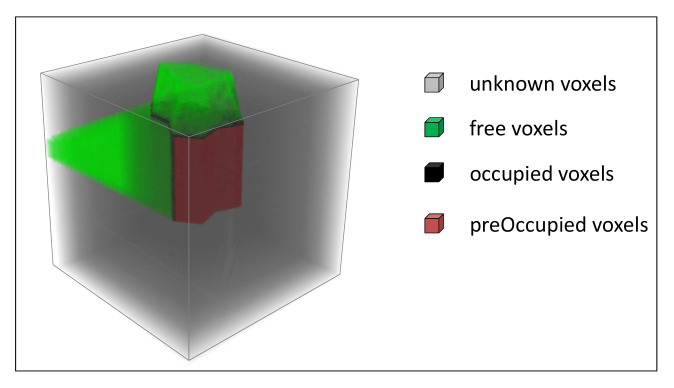
Voxels’ status in the model space.

**Figure 6 entropy-24-00578-f006:**
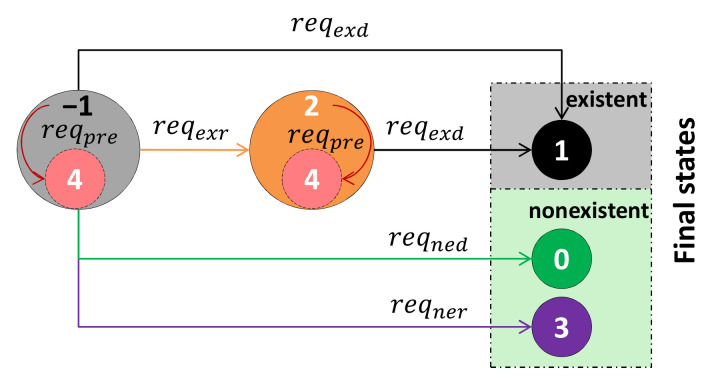
Transformation between the existence states of a feature.

**Figure 7 entropy-24-00578-f007:**
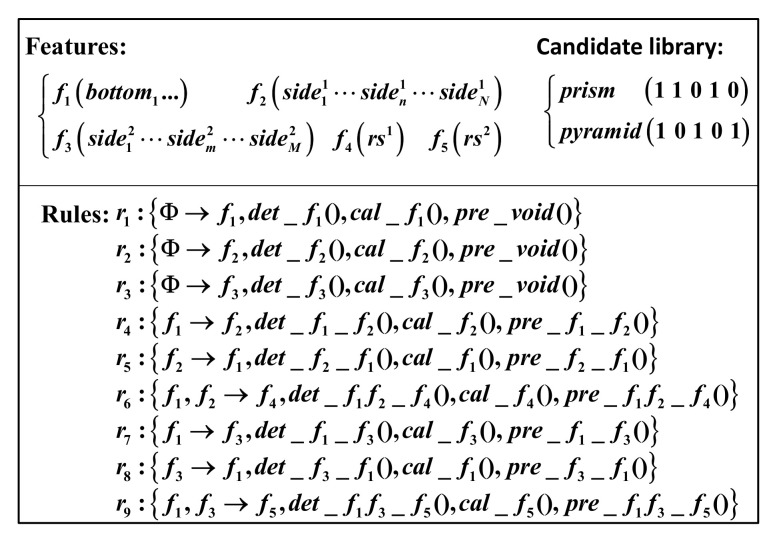
Prior knowledge library of the objects for reconstruction.

**Figure 8 entropy-24-00578-f008:**
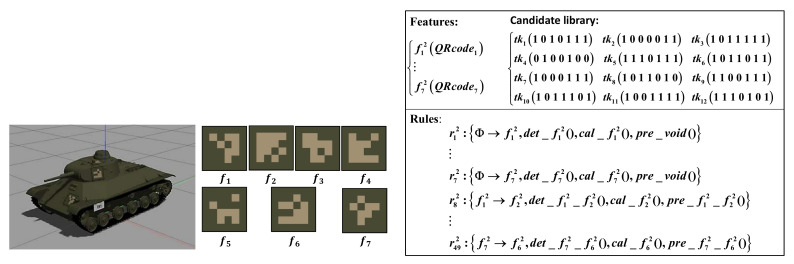
Prior knowledge library of the tanks for recognition.

**Figure 9 entropy-24-00578-f009:**
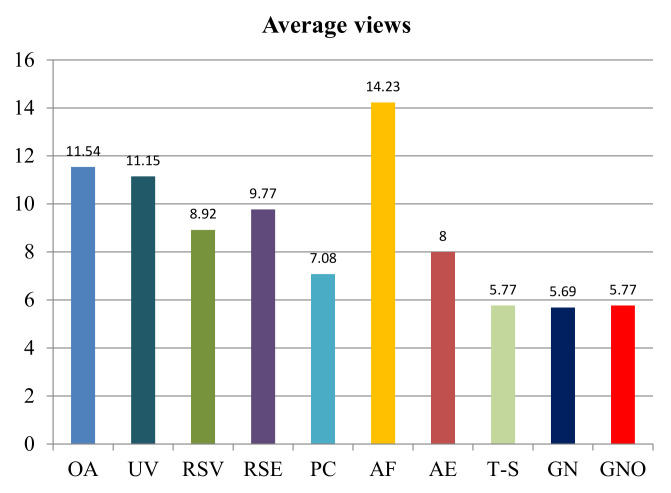
Average number of views needed by each method in reconstruction experiments.

**Figure 10 entropy-24-00578-f010:**
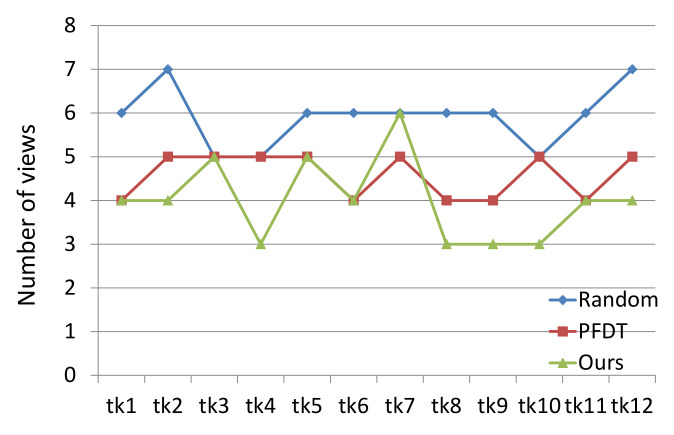
Number of views needed in recognition experiments.

**Table 1 entropy-24-00578-t001:** Comparison of each model for reconstruction. T-S, GN and GNO achieve better performance.

Models	m1	m2	m3	m4	m5	m6	m7	m8	m9	m10	m11	m12	m13
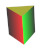	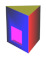	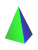	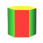	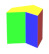	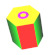	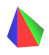	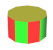	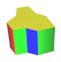	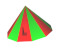	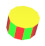	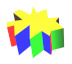	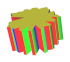
OA	8	20	20	8	8	8	5	9	9	20	10	8	17
UV	8	20	13	8	8	8	5	9	9	20	14	8	15
RSV	6	17	4	5	9	6	4	16	8	**5**	11	13	12
RSE	6	16	6	**4**	8	**4**	4	14	12	11	8	14	20
PC	5	9	5	5	**5**	5	3	6	**7**	8	5	16	13
AF	9	10	5	20	17	20	18	7	20	10	9	20	20
AE	**4**	8	7	6	6	8	5	6	8	10	7	14	15
T-S	5	**7**	4	5	**5**	**4**	5	6	**7**	**5**	**4**	8	10
GN	5	8	**3**	5	**5**	5	3	**5**	**7**	7	5	9	**7**
GNO	8	8	**3**	6	**5**	6	**2**	6	8	**5**	5	**6**	**7**

## Data Availability

The dataset of this paper can be found at https://github.com/Miss-Katherine/DAE-model-of-prisms, accessed on 17 April 2022.

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
