# Peer review of "A Generic View Planning System Based on Formal Expression of Perception Tasks†"

_entropy, 2022, doi:10.3390/e24050578_

Round 1
Reviewer 1 Report
This paper presents a generic view planning system based on formal expressions of perception tasks. The authors made experiments for view planning for 3D recognition and reconstruction and they achieved good performance and results on different tasks. Its a nice research paper. The authors should add more references and the information given in introduction with related works sections should be given or discussed in more details.
Reviewer 2 Report
This paper introduces a novel generic view planning method a technique that guides adjustment of sensor positions in multi-view "perception" (visual sensing) tasks on the basis of active "perception" (sensing). The method aims for "intelligent" view planning by reducing resource consumption of robots designed for multiple agencies (tasks) involving the visual processing of shape information. Formal task descriptions are programmed into the system in terms of libraries that contain prior information relative to objects and their visual representation status or representation mode (the authors call this "perception" without defining what is to be understood by "perception here in this context - I will get back to this point later with a clear recommendation), and the task completion (task state) status. These prior data form the basis of View Planning (VP) for the execution of a specific task. Any task begins with the initialization of state descriptions, and states are updated after a novel measurement where the "next best task space view" is calculated by the system until the task is ended by a signal that the completion status has attained a target state. Simulation experiments for system abilities for reconstruction and recognition were performed. Results suggest that the proposed algorithms permit effective VP for multi-task robotic agency. The limitations of the approach are clearl identified by pointing out that the parameters of the computations can only deal with a limited degree of complexity of local surfaces and their shape characteristics. This limitation related to visual object complexity is expected to be overcome by building deep learning into the system.
This is a good paper presenting a novel framework for the control of robotic agency by view planning in sensor driven multitasking involving the visual processing of variable shape positions. The manuscript is clearly written. Advantages and limitations are pointed out, but are not discussed well enough for the reader to understand a) the full importance of this method and b) its full potential for future development. The following major revisions of the manuscript text are mandatory to fix this:
- ) The term "perception" refers to a complex definition, that the authors do not provide. What is referred to here in the paper is sensor-driven visual representation for robotic action. I recommed replacing "perception" by "visual representation" everywhere in the text to make this quite clear.
- ) From this follows the problem of external validity of the method. The experimental simulations here only provide a level of internal validity by assessing algorithm performance for a specific set of robotic tasks. No suggestions are made how external validity could be assessed by experiments on perception (i.e. human perception for bioinspired robots).
- ) The suggestion that the limitations of the method (which is not capable of dealing with complex shape and surface properties) will be overcome by building "deep learning" into the system appears naive. What kind of "deep learning"? Given the level of robotic agency tackled here, it is quite clear that bioinspired unsupervised learning (Hebbian synapse- based reinforcement learning) will a) provide a solution to the problem of shape complexity and b) allow for implementing aspects of external validity by enabling experiments on perception (human) beyond mere simulations of algorithm performance.Conclusion: 1) needs to be implemented as stated 2) and 3) needs to be addressed explicitly in the discussion section of the manuscript including the two most recent, mandatory references:
- Wan, C.; Cai, P.; Guo, X.; Wang, M.; Matsuhisa, N.; Yang, L.; Lv, Z.; Luo, Y.; Loh, X.J.; Chen, X. An artificial sensory neuron with visual-haptic fusion. Nature Communications. 2020, 11, e4602.
- Dresp-Langley, B. From biological synapses to “intelligent” robots. Electronics. 2022, 11(5), e707.
Round 2
Reviewer 2 Report
The authors have revised their manuscript carefully and to full satisfaction, describing their outstanding work and all the subtle detials and implications very well. The paper makes an important contribution to this field, and opens novel perspectives for groundbreaking future research. The Reviewer warmly recommends the article for publication in its present form.